# Epistaxis and thrombocytopenia as major presentations of louse borne relapsing fever: Hospital-based study

**Eyob Girma Abera**[1,2☯]*, **Kedir Negesso Tukeni**[3], **Gelaw Hailemariam Didu**[4], **Temesgen Kabeta Chala**[5], **Daniel Yilma**[2,3], **Esayas Kebede Gudina**[2,3☯]

1 Department of Public Health, Jimma University, Jimma, Oromia, Ethiopia, 2 Clinical Trial Unit, Jimma University, Jimma, Oromia, Ethiopia, 3 Department of Internal Medicine, Jimma University, Jimma, Oromia, Ethiopia, 4 Department of Emergency Medicine, Jimma Medical Center, Jimma, Oromia, Ethiopia, 5 Department of Health Policy and Management, Jimma University, Jimma, Oromia, Ethiopia

☯ These authors contributed equally to this work.
* eyob.girma@ju.edu.et

## Abstract

### Background

Louse-borne relapsing fever (LBRF) remains a cause of sporadic illness and occasional outbreaks in Ethiopia and other east African countries in overcrowded and unhygienic settings. In this article, we present clinical profiles and treatment outcome of patients treated as confirmed or probable cases of LBRF at Jimma Medical Center (JMC) in southwest Ethiopia.

### Methods

Patients treated as confirmed or probable cases of LBRF at JMC during a period of May–July 2022 were prospectively followed during their course of hospital stay. All patients were evaluated with blood film for hemoparasites, complete blood count, and liver enzymes on hospital presentation. They were followed with daily clinical evaluation during their hospital stay.

### Result

Thirty-six patients were treated as cases of LBRF. All patients except one were from Jimma Main Prison in Jimma Town, Ethiopia. All the patients were male with mean age of 28.7 years (SD = 12.7). The diagnosis of LBRF was confirmed by detection of *B. recurrentis* in blood film of 14 (38.9%) of the patients; the rest were treated as presumptive case of LBRF. Fever, reported by all patients, and an acute onset epistaxis, 30 (83.3%), were the major reasons for healthcare visits. Twenty-two (61.1%) patients were having thrombocytopenia with a platelet count < 150,000/μL; nine (25%) of which had severe forms (<50,000/μL). All patients were treated with oral doxycycline and discharged with improvement after a mean length of hospital stay of 4.25 days (SD = 0.77), range 2–6 days. Public health emergency was activated within two days of the first cases and helped in delousing all the cases and their contacts.

**Data Availability Statement:** All relevant data are within the manuscript.

**Funding:** The authors received no specific funding for this work.

**Competing interests:** The authors have declared that no competing interests exist.

## Conclusion

LBRF remains a public health problem in Ethiopia in settings with poor personal hygiene. Patients with LBRF may present with severe thrombocytopenia and life-threatening bleeding. Early detection and treatment initiation prevents outbreak propagation and improves treatment outcome.

## Background

Louse-borne relapsing fever (LBRF) is a vector borne ancient epidemic disease caused by *Borrelia recurrentis*, with descriptions dating back to Hippocrates' times [1]. From 1903 to1936, an estimated 50 million cases with 10% mortality were recorded across the Middle East, Northern and Eastern Africa [2]. Furthermore, a second epidemic occurred during 1943–46 with 10 million cases [2]. Twenty thousand cases with 10% death rate were reported among Dinka tribe in South Sudan during 1998–1999 [3]. The last few decades, reports of LBRF were almost exclusively limited to the Horn of Africa [4]. However, in July 2015, LBRF has been diagnosed in almost 100 young male refugees who sought asylum in European countries [2, 5]. In Ethiopia, the biggest outbreak was recorded in 1991 after the civil war in Asella where 389 people were infected with a hospital fatality rate of 3.6% [6]. During 1997–2001, Jimma Hospital reported 617 LBRF cases [7].

There are several factors that promote LBRF outbreaks including, war, forced migrations, prisons, poverty, famine, poor personal hygiene and cold or rainy season. Person to person transmission of *B. recurrentis* is restricted to one vector, the human body louse *Pediculus humanus corporis* that retreats from the skin after feeding to hide and lay their eggs in clothing [2].

LBRF is typically characterized by the acute onset of febrile disease and subsequent febrile relapses with the incubation period of 4–18 days (average of 7 days) [2]. Ninety percent of patients presented with fever accompanied by headache, myalgia, arthralgia, nausea, abdominal pain, sweating and tachycardia, and is less frequently by hepatomegaly (17–66%), splenomegaly (41–77%), petechial rash (8–28%), respiratory systems (16–34%), jaundice (7–36%) and epistaxis [8, 9].

Diagnosis is confirmed by detecting spirochetes in Giemsa stained peripheral thin and thick blood smear of febrile patients [2, 10]. The sensitivity of a blood smear is approximately 70% in febrile patients but decreases to less than 5% during an afebrile period [11]. Due to low level of spirochetes in the blood, and the spirochetes' thin and transparent morphology, it is difficult to detect by ordinary light microscopy. Therefore, a quantitative buffy coat (QBC) technique has been reported as a sensitive method of detecting spirochetes [12–14]. Polymerase chain reaction (PCR) is a highly effective test in detecting and identifying *Borrelia* species with 100% sensitivity and specificity for *B. recurrentis* [2, 9]. However, PCR detects the infection only at the febrile stage and not during the incubation period prior to the onset of clinical symptoms [15]. Serologic test are used as alternative diagnostic method, but shows limited specificity due to cross-reactivity among *Borrelia* species and the complexity of the relapsing phenomenon [4, 9].

A single dose of erythromycin, tetracycline, and doxycycline (100mg orally every 12 hours) are the choices of antibiotics for LBRF [8, 10]. Within two hours of antibiotics therapy, the release of inflammatory contents resulting from bacterial lysis may lead to a flu-like response which is called Jarisch-Herxheimer reaction (JHR). The mortality rate of the LBRF is 10–40%

for untreated and below 5% for treated patients [8]. Up to eight relapses have been observed in patients with LBRF despite most infected patients experience up to two relapses. The relapsing pattern is due to the antigenic variation of *Borrelia* organisms lipoproteins in the blood [4, 11, 16].

In Ethiopia, due to the prevalence of malaria and other acute febrile illness with their non-specific sign and symptoms, RF may be misdiagnosed and underreported. In this case series study, we assessed the clinical presentation of 36 confirmed or probable cases of LBRF treated at Jimma Medical Center (JMC) during a period of May to July 2022.

## Methods

### Study design and setting

We prospectively followed all cases treated as confirmed or probable cases of relapsing fever at Jimma Medical Center in Jimma Town, southwest Ethiopia during months of May to July 2022. Jimma Town is one of the biggest town in southwest Ethiopia with estimated population of 207,573 in 2021 [17]. Jimma Medical Center is a tertiary teaching hospital located in Jimma Town serving a population of over 20 million [18].

### The cases

All blood film confirmed or clinical suspected cases of LBRF treated at Jimma Medical Center during Months of May to July 2022 were included. Confirmed RF was defined if *Borrelia* species was detected in Giemsa-stained thick and thin blood film. Suspected RF was defined and treated as probable cases if patients presented with clinical presentations suggestive of RF and had a known contact with confirmed RF patient; who otherwise had negative BF for *Borrelia* species and there was no other medical condition explaining their presentation.

### Data collection

A structured case report form was used to collect the data prospectively. The Data on socio-demographic characteristics, clinical presentations, laboratory profile, treatments, and discharge outcomes were collected.

### Data processing and analysis

The collected data were checked for completeness, and then double entered, validated, and cleaned using EpiData version 3.1 before it was exported to SPSS® version 25 (IBM®, New York, USA) for analysis.

### Ethical approval

Ethical clearance was obtained from the Institutional Review Board (IRB) of Jimma University, Institute of Health. Written informed consent from the study participants and an official letter of permission from Jimma Main Prison were obtained. The collected data was kept confidential through anonymity.

## Result

Thirty-six patients were treated as relapsing fever; 14 (38.9%) were blood film (BF) confirmed case while 22 (61.1%) were treated as probable cases of relapsing fever. All of the patients, except one, were from Jimma Main Prison in Jimma Town. All were men with a mean age of 28.7 years (SD = 12.7); range of 16 years to 70 years. Thirty-one of the patients, all from the

prison, were treated in May 2022 during the first two weeks of the outbreak. The other five patients presented as sporadic cases over the next two months.

## Clinical characteristics

Fever was reported as a major presenting symptom by all of the patients. Headache and joint pain were other common presenting symptoms reported in 97.2% (n = 35) of the patients each. Acute onset nasal bleeding was reported by 83.3% (n = 30) of patients and was the main reason for healthcare visit. Fever (axillary temperature of >37.5˚C) and tachycardia (heart rate >100bpm) were the common findings on physical examination (Table 1).

## Laboratory profile of the cases

Blood film was performed for all participants on presentation and *Borrelia* species was detected in blood film of 14 (38.9%) of the cases. However, there was no difference in the clinical presentation of patients who had positive blood film and negative findings. Complete blood count (CBC) was performed for all patients. Leukocytosis defined as peripheral white blood cell count (WBC) of >11,000/μl [19] was reported in nine patients (25%), while 13 patients (36.1%) had anemia (hemoglobin <13.5g/dL). Chest X-ray was performed for two patients who reported respiratory symptoms. A finding consistent with post-tuberculosis lung diseases was reported in the one and the other patient was with normal finding (Table 2).

## Thrombocytopenia and epistaxis

Thrombocytopenia (platelet count < 150,000/μL) was reported in 22 (61.1%) patients; nine (25%) of them had severe thrombocytopenia (<50,000/μL). Twenty patients with thrombocytopenia (90.1%) and eight of those with severe thrombocytopenia presented with epistaxis. However, no statistically significant association between thrombocytopenia and epistaxis was observed (*P-value = 0.304*) (**Fig 1**).

## Case management and outbreak control

Based on clinical manifestations and epidemiological background, all patients with confirmed and probable cases of LBRF were managed in the isolation unit. Doxycycline 100 mg twice

**Table 1. Clinical presentation of patients treated as LBRF at Jimma Medical Center, Ethiopia, May–July 2022.**

| Cases and Symptoms | Frequency | Percentage (%) |
|---|---|---|
| Fever (self-reported) | 36 | 100 |
| Headache | 35 | 97.2 |
| Joint pain | 35 | 97.2 |
| Easy fatigability | 32 | 88.9 |
| Nasal bleeding | 30 | 83.3 |
| Body ache | 29 | 80.6 |
| Poor appetite | 25 | 69.4 |
| Cough | 5 | 13.9 |
| Vomiting | 4 | 11.1 |
| Diarrhea | 3 | 8.3 |
| Abdominal cramp pain | 2 | 5.6 |
| Skin rash | 1 | 2.8 |
| Fever (Axillary body temperature(>37.5˚C)) | 29 | 80.6 |
| Tachycardia (>100 beats/minutes) | 29 | 80.6 |

**Table 2. Laboratory characteristics of patients treated as LBRF at Jimma Medical Center, Ethiopia, May–July 2022.**

| Laboratory Test | N* | Normal (%) | High (%) | Low (%) |
|---|---|---|---|---|
| White blood cell count (WBC)[a] | 36 | 25(69.4) | 9(25) | 2(5.6) |
| Hemoglobin (Hgb)[b] | 36 | 21(58.3) | 1(2.8) | 14(38.9) |
| Platelet[c] | 36 | 14(38.9) | 0 | 22(61.1) |
| Aspartate aminotransferase (AST)[d] | 33 | 19(57.5) | 12(36.4) | 2(6.1) |
| Alanine aminotransferase (ALT)[e] | 34 | 33(97.1) | 1(2.9) | 0 |
| Alkaline phosphatase (ALP)[f] | 31 | 29(93.5) | 0 | 2(6.5) |
| Prothrombin Time (PT)[g] | 8 | 0 | 2(25) | 6(75) |
| Partial Thromboplastin time (PTT)[h] | 8 | 1(12.5) | 0 | 7(87.5) |
| Iternational Normalized Ratio (INR)[i] | 8 | 5(62.5) | 1(12.5) | 2(25) |
| Total bilirubin[k] | 31 | 29(93.5) | 2(6.5) | 0 |
| Direct bilirubin[j] | 31 | 22(70.9) | 9(29.1) | 0 |
| Indirect bilirubin[l] | 29 | 23(79.3) | 6(20.7) | 0 |

**a. WBC;** High $\geq$11,000/μl, Low <4,000/μl.

**b. Hgb;** High >16g/dl, Low <13.5g/dl.

**c. Platelet;** High>450,000/μl, Low <150,000/μl.

**d. AST;** High >33U/ L, LOW <8U/L.

**e. ALT;** High >55U/L, Low<7U/L.

**f. ALP;** High>157U/L, Low <44 U/L.

**g. PT**; High >13.5 sec, Low<11 sec.

**h. PTT**; High >35 sec, Low<25sec.

**i. INR**; High >1.1, Low<0.8.

**j. Direct bilirubin**; Normal <0.3mg/dl.

**k. Total bilirubin**; High >1.2mg/dl, Low<0.1mg/dl.

**l. Indirect bilirubin**; High $\geq$0.8mg/dl, Low<0.2 mg/dl.

* Number of patients with specific laboratory data.

daily, paracetamol 1 gm at presentation, and intravenous fluid were given for all patients. One patient who presented with an active nasal bleeding, severe thrombocytopenia, and severe anemia was transfused with one unit of whole blood. One known hypertensive patient was treated additionally with hydrochlorothiazide. All patients recovered after commencement of treatment without fever recrudescence or occurrence of shock, any JHR or other complication. All patients were discharged with improvement. The mean length of hospital stay was 4.25 days (SD = 0.77), range 2–6 days. All the patients and their known contacts were deloused.

Besides pharmacological treatments, public health emergency measures were activated at the prison and its vicinity to actively screen contacts for symptoms and early treatment as well environmental cleaning such as washing of infested clothes with boiled water. All patients and inmates sharing the same cells as cases were bathed with soap, their head hair was shaved, pesticides spray was applied in the prison and health education was provided.

## Discussion

Cluster of cases of louse borne relapsing fever from Jimma Town Main Prison in Ethiopia were presented with major complaints of fever and epistaxis in May 2022. Public emergency was activated immediately, and the index cases and their close contacts were evaluated and treated promptly. As a result, most of the outbreak was contained within the prison and just one of the cases was reported from the community. Although sporadic cases were reported until July 2022, most of the cases (n = 31) were treated within two weeks of the outbreak onset.

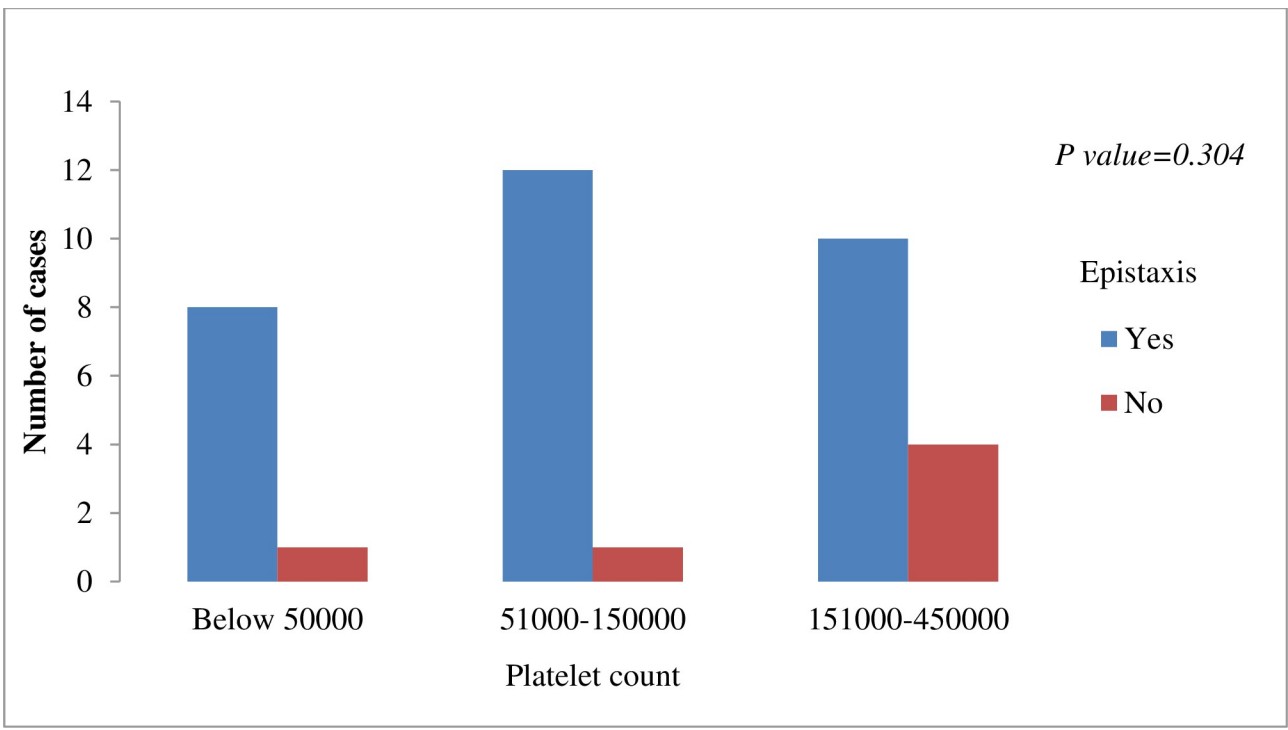

**Fig 1. Thrombocytopenia and epistaxis in patients treated as LBRF at Jimma Medical Center, Ethiopia, May–July 2022.**

Thrombocytopenia was reported in most of the cases with a quarter of them having severe forms. All patients were successfully treated with fluid and antibiotics and discharged with no major sequelae.

Although recent evidences have shown a gradual decline of endemicity of louse borne relapsing fever in Ethiopia [20], outbreaks have been reported in crowded and unhygienic settings such as prisons and refugee camps [6, 21, 22]. Only few sporadic cases of LBRF were treated at the hospital for the past two decades. However, due to the disease's presentation with non-specific symptoms [23] and huge overlap with malaria symptoms [24], it is highly possible that cases might have been missed. The current onset of the outbreak indicates that overcrowding, poor socioeconomic status and/or poor personal hygiene are important risk factors for occurrence of LBRF in Ethiopia [8]. It also shows that, LBRF remains an important public health problem in Ethiopia despite improved healthcare system over the past few decades. Furthermore, due to limited diagnostic options and the similarity of sign and symptoms with other febrile illness, particularly, malaria, and COVID-19, diagnosis and early detection of RF cases may be a challenge in Ethiopia. This may lead to undetected sporadic cases which may escalate to full-blown outbreak in crowded settings.

Most infected individuals develop symptoms within 5 to 15 days of the infection. LBRF is known for cycle of symptoms that may come and go based on the level of spirochetes in the blood. Fever, headache and joint/muscle aches are major presenting symptoms [10]. Although not described as a major presenting symptom of LBRF, bleeding diathesis associated with low platelet count has been reported before [25]. The fact that epistaxis, along with fever, was the major reasons why patients sought care and that a quarter of the patients presented with severe and life-threatening thrombocytopenia is not congruent with what have often been reflected in most previous reports. Although describing the pathophysiologic mechanism is beyond the

scope this study, we can speculate that the patients presented after repeated cycles of spirochetemia that might have ultimately led to severe thrombocytopenia and bleeding diathesis. It has also been reported in the previous study that the degree of thrombocytopenia and bleeding are associated with the duration of the illness [25]. It is thus possible to assume that patients had several cycles of spirochetemia and fever relapses before the current presentation even though the onset of the current presentation was reported to be only few days. It also possible that such complications might have been underreported in previous studies. Relapsing fever affects the most underdeveloped part of the world and confined to the most neglected society [2] and remains one of the least investigated diseases. As a result, its pathophysiology and optimal management have never been studied extensively.

Microscopic visualization of spirochetes on dark/bright field examination of Wright/ Giemsa stained thick and thin blood smears during febrile episode with recurrent fever is a confirmatory diagnosis of relapsing fever [8, 26]. However, the spirochetes are too scanty to be visualized in light microscope when smears are performed during afebrile episodes and should be repeated when the fever reappears [27]. As a result, the sensitivity of a blood smear is approximately 70% in febrile patients but decreases to less than 5% during an afebrile period [11, 27]. In this study, only 14 patients (38.9%) were confirmed positive for *B. recurrentis* with a microscopic evaluation. The rest of 22 patients (61.1%) were probable cases with similar sign and symptoms, but negative blood film, which may be due to the low level of spirochetes in the blood and/or the thin and transparent morphology of the spirochetes that makes the detection difficult under light microscope [12]. Due to its low sensitivity, this diagnostic method has been replaced by PCR [28]. However, because of the unavailability PCR, the diagnosis of relapsing fever in Ethiopia and most affected areas dependent on blood smear examination. Hence, proactive approach is needed in the care of LBRF in outbreak settings like in this case where some of the patients had blood film confirmed spirochetemia. It may thus be pragmatic to treat symptomatic patients who had clear contacts with confirmed cases.

As symptoms of relapsing fever are non-specific and overlap with other febrile infections such as malaria, which also shares similar geographic distribution, empiric treatment of LBRF may come at cost of irrational antibiotic use and misdiagnosis of other diseases. Thus, a rational approach in the management of patients suspected to have relapsing fever should put those facts into consideration. All the 22 patients who had negative blood film for spirochetes in this case series were treated as presumptive cases of LBRF while the possibility of malaria has been ruled out with repeated blood film examination. Moreover, all of them had clinical response within two days of treatment commencement and were discharged with improving suggesting the possibility of LBRF.

Patients with other medical conditions were also treated accordingly. One patient with known hypertension case was managed by hydrochlorothiazide, triple therapy (Clarithromycin, Amoxicillin and Omeprazole) was prescribed for one patient with peptic ulcer disease, a 21 year old patient with low hemoglobin (10.9g/dL) and very low platelets (19,000/μl) count was transfused with a whole blood as one unit of fresh whole blood increased a platelet count and mean platelet volume to a level higher than that achieved by 6 and 10 platelet units respectively [29]. After three days of admission, the patient's platelet count increased to 162,000/μl although the hemoglobin level decreased to 9.8g/dL.

Within two hours of antibiotics therapy, the release of inflammatory contents from the bacteria lysis may lead to a flu-like response called Jarisch-Herxheimer reaction (JHR) especially following penicillin treatment [8, 30]. Hence, the patients should be strictly followed and given paracetamol before and after 2 hours of antibiotic commencement to minimize the severity of JHR [8, 31]. As JHR incidence rate in LBRF is 55.8% [32], one would expect such cases in this study. However, there were no reported JHR cases. This could be due to underreporting or

under diagnosis because of poor follow up after the initiation of antibiotics treatment. In a meta-analysis of six studies in Ethiopia of patients with LBRF, JHR occurred in 89 of 239 patients (37%) treated with penicillin and in 96 of 199 patients (48%) treated with tetracycline [33]. Myocarditis is another complication commonly reported in LBRF patients [2] but not reported in this case series. As with JHR, this can be due to lack of proper evaluation for these complications. Nevertheless, none of the patients reported symptoms related to these complications. All the patients were discharged with improvement indicating that none developed any serious complication. This may be due to the fact that almost all of the affected patients were young adults with no major underlying medical conditions. Additionally, all of them were treated promptly with antibiotics, IV fluids, and paracetamol that might have led to early recovery without major complications.

## Strengths and limitations

This case series would help complement the evidence gap in louse borne relapsing fever. It highlights the importance of high index of suspicion for the diagnosis and treatment of LBRF in "high-risk settings". Moreover, the presentations of epistaxis and thrombocytopenia have also been highlighted as the important presentations and complications in patients with LBRF. However, it lacks exhaustive evaluation of patients' clinical course in the hospital. Besides, most of the patients were treated without confirming the diagnosis. ECG and echocardiographic evaluation to detect myocarditis, an important complication, were not performed.

## Conclusions

Louse borne relapsing fever remains an important public health problem in Ethiopia. Patients with LBRF may present with severe thrombopenia and bleeding diathesis. Giemsa-stained blood film remains the mainstay of relapsing fever diagnosis in Ethiopia despite its low yield. Hence, treatment decision should additionally rely on clinical presentation, epidemiologic patterns, and laboratory tests to rule out diseases with similar presentation. Early case detection and treatment initiation prevent outbreak propagation and improve treatment outcome.

## Acknowledgments

We would like to express our sincere and deep-rooted thanks to the Jimma Emergency Operation Center, Jimma Medical Center, Jimma Main Prison, Ethiopian Public Health Institute, Oromia Regional Health Bureau, Jimma Zone Health Department, Jimma Town Health Department and study teams, for the support provided during intervention and data collection.

## Author Contributions

**Conceptualization:** Daniel Yilma, Esayas Kebede Gudina.

**Data curation:** Kedir Negesso Tukeni, Gelaw Hailemariam Didu, Temesgen Kabeta Chala.

**Formal analysis:** Temesgen Kabeta Chala.

**Investigation:** Kedir Negesso Tukeni, Gelaw Hailemariam Didu.

**Methodology:** Esayas Kebede Gudina.

**Project administration:** Esayas Kebede Gudina.

**Resources:** Gelaw Hailemariam Didu.

**Software:** Eyob Girma Abera.

**Supervision:** Esayas Kebede Gudina.

**Validation:** Daniel Yilma.

**Writing – original draft:** Eyob Girma Abera.

**Writing – review & editing:** Eyob Girma Abera, Daniel Yilma, Esayas Kebede Gudina.

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
