## [Decision Letter · Decision Letter 0]

6 Dec 2022

PONE-D-22-29426Epistaxis and thrombocytopenia as major presentations of louse borne relapsing fever: hospital-based studyPLOS ONE

Dear Dr. Abera,

Thank you for submitting your manuscript to PLOS ONE. After careful consideration, we feel that it has merit but does not fully meet PLOS ONE’s publication criteria as it currently stands. Therefore, we invite you to submit a revised version of the manuscript that addresses each of the points raised during the review process. I apologize for the slowness of this review. I reached out to numerous experts in the field, but only 1 accepted the invitation and submitted a review. I also read your manuscript and concur with the reviewer's comments. Rather than delay this further, I am sharing the reviewer's comments with you, so that you can respond.

We look forward to receiving your revised manuscript.

Kind regards,

Brian Stevenson, Ph.D.

Academic Editor

PLOS ONE

Journal Requirements:

3. We note you have included a table to which you do not refer in the text of your manuscript. Please ensure that you refer to Table 2 in your text; if accepted, production will need this reference to link the reader to the Table.

Reviewers' comments:

Reviewer's Responses to Questions

**Comments to the Author**

1. Is the manuscript technically sound, and do the data support the conclusions?

Reviewer #1: Yes

2. Has the statistical analysis been performed appropriately and rigorously? 

Reviewer #1: N/A

3. Have the authors made all data underlying the findings in their manuscript fully available?

Reviewer #1: Yes

4. Is the manuscript presented in an intelligible fashion and written in standard English?

Reviewer #1: No

5. Review Comments to the Author

Reviewer #1: This study reports an outbreak of louse-borne relapsing fever (LBRF) in Ethiopia. There were 36 sick patients and spirochetes were visualized in blood smears of 14. The remaining patients were presumed to be infected with LBRF spirochetes basked on proximity to the positive patients. This study demonstrates the continued impact of LBRF in Ethiopia. This reviewer thought the information is needed and timely because of the neglected nature of relapsing fever. A limitation of the study was that the authors’ never confirmed whether the presumed cases of relapsing fever were indeed true positives. However, they do address this limitation in the study. The primary concern for this reviewer was the presentation of the manuscript, as outlined below.

1. Lines 59-63: This reviewer suggests adding a topic sentence, so the paragraph is not a two-sentence paragraph. For example, the authors could add the following topic sentence, “There are several factors that promote LBRF outbreaks.”

2. Lines 64-79. These two paragraphs could be combined into one.

3. Lines 744-75. The sentence needs editing. It is written, “Therefore, a quantitative buffy coat (QBC) technique has been reported a sensitive…” This reviewer thinks the authors mean “as a sensitive” I underlined “as” to point out that I believe it is a missing word.

4. Lines 75-77. This sentence should be reworded. In particular, the “100% sensitivity and specificity.” Context is needed. For example, when B. recurrentis has been cleared by the host antibody response, PCR will not detect spirochete DNA in the blood. Did the studies in reference 2 and 9 demonstrate detection of Borrelia DNA between spirochetemic episodes?

5. Line 118. The meaning of BF should be written out the first place it appears in the manuscript.

6. Lines 138-140. The sentence beginning with “Chest X-ray…” could be broken up into two sentences. The second sentence should start at, “Findings consistent…”

7. Lines 161-162. This sentence reads awkwardly and needs editing. A suggestion is

8. This reviewer does not think figure 1 is needed.

9. Lines 168-170. As written, this is not a sentence. Please correct.

10. Lines 170-172. This sentence is difficult to understand. Please correct.

11. Line 173. “due their unavailability” is incorrectly written. Also, “their” is not the correct word to use. This sentence may not be needed and could be omitted from the manuscript.

12. Line 174. For more clarity, the sentence “All patients had smooth in hospital course after commencement of treatment….” could be written, “All patients recovered after the commencement of treatment….”

13. Liners 177-179. For simplicity, the sentence could be written, “All the patients and their known contacts were deloused.” Also, this sentence should be the topic sentence for the following paragraph.

14. Line 186. “Louse” does not need to be capitalized.

15. Line 205. There needs to be a comma after malaria.

16. Lines 203-206. This reviewer suggests deleting the words “which are causing occasional outbreaks in the country now”

17. Line 234. “thus” should be changed to “which”

18. Line 243. A comma is needed before “which”.

19. Line 251. For clarity, a period should be placed after “accordingly” and the next sentence should read, “One patient with known ….”

20. Line 271. “Besides” should be deleted.

21. Lines 277-280. “susception” is confusing. Please edit this sentence. Also, in line 279, “bleeding and thrombocytopenia as important presentations….” could be reworded “We reported that bleeding and thrombocytopenia are important presentations and complications.” Overall, the sentences in lines 277-283 are difficult to understand. Editing is needed.

6. PLOS authors have the option to publish the peer review history of their article (what does this mean?). If published, this will include your full peer review and any attached files.

Reviewer #1: No

---

## [Author Response · Author response to Decision Letter 0]

8 Dec 2022

We would like to thank the editor and reviewer for their comments and suggestions to make the paper valuable. Kindly, we have updated the manuscript as per the comments and suggestions.

---

## [Editor Report · Decision Letter 1]

13 Dec 2022

Epistaxis and thrombocytopenia as major presentations of louse borne relapsing fever: hospital-based study

PONE-D-22-29426R1

Dear Dr. Abera,

We’re pleased to inform you that your manuscript has been judged scientifically suitable for publication and will be formally accepted for publication once it meets all outstanding technical requirements.

Kind regards,

Brian Stevenson, Ph.D.

Academic Editor

PLOS ONE
---

## [Editor Report · Acceptance letter]

20 Dec 2022

PONE-D-22-29426R1 

Epistaxis and thrombocytopenia as major presentations of louse borne relapsing fever: hospital-based study 

Dear Dr. Abera:

I'm pleased to inform you that your manuscript has been deemed suitable for publication in PLOS ONE. Congratulations! Your manuscript is now with our production department. 

Kind regards, 

on behalf of

Prof. Brian Stevenson 

Academic Editor

PLOS ONE